# Analysis of the Lack of Follow-Up of Bariatric Surgery Patients: Experience of a Reference Center

**DOI:** 10.3390/jcm11216310

**Published:** 2022-10-26

**Authors:** Marie Auge, Olivier Dejardin, Benjamin Menahem, Adrien Lee Bion, Véronique Savey, Guy Launoy, Véronique Bouvier, Arnaud Alves

**Affiliations:** 1Department of Digestive Surgery, University Hospital of Caen, CEDEX 9, 14033 Caen, France; 2L’Unité Anticipe (Inserm U1086), CEDEX 5, 14076 Caen, France; 3Department of Research, University Hospital of Caen, CEDEX 9, 14033 Caen, France; 4Registre des Tumeurs Digestives du Calvados, CEDEX 5, 14076 Caen, France

**Keywords:** bariatric surgery, weight loss, follow-up

## Abstract

Few studies have evaluated the association between non-clinical and clinical determinants in terms of discontinuing follow-up after bariatric surgery. This cohort study aims to assess these associations. Data were collected from a prospectively maintained database of patients who underwent laparoscopic bariatric surgery from January 2012 to December 2019. The Cox model was used to assess the influence of preoperative determinants on follow-up interruptions for more than one year. Multilevel logistic regression was used to evaluate the association between clinical factors and post-operative weight loss with the regularity of follow-up. During the study period, 9607 consultations were performed on 1549 patients. The factors associated with a follow-up interruption from more than 365 days included male gender (HR = 1.323; CI = 1.146–1.527; *p* = 0.001) and more recent years of intervention (HR = 1.043; CI = 1.012–1.076; *p* = 0.0068). Revisional bariatric surgery was associated with a lower risk of follow-up interruption (HR = 0.753; CI = 0.619–0.916; *p* = 0.0045). Independent risk factors of an irregular follow up were higher age (HR = 1.01; CI = 1.002–1.017; *p* = 0.0086); male gender (OR = 1.272; CI = 1.047–1.545; *p* = 0.0153); and higher %TWL (Total Weight Loss) (OR = 1.040 CI = 1.033–1.048 *p* < 0.0001). A higher preoperative BMI (OR = 0.985; CI = 0.972–0.998; *p* = 0.0263) and revisional surgery (OR = 0.707; CI = 0.543–0.922; *p* = 0.0106) were protective factors of irregularity. This study suggests that the male gender and most recent dates of surgery are the two independent risk factors for follow-up interruption. Older age, male gender, and higher weight loss were all independent risk factors of an irregular follow-up. Revision bariatric surgery is a protective factor against interruption and irregular follow-up with a higher preoperative BMI. Further studies are needed to obtain long-term results in these patients with discontinued follow-ups.

## 1. Introduction

Affecting more than 13% of the world population, obesity is a major public health problem [1]. In France, overweight and obesity affect nearly half of the population, according to the data of the Obepi study published in 2020 [2].

Bariatric surgery is considered the most effective treatment for severe obesity in long-term weight loss and the resolution of obesity-related comorbidities (i.e., diabetes, arterial hypertension, dyslipidemia, and obstructive sleep apnea syndrome OSAS) [3]. It is also associated with a significant reduction in the incidence of obesity-related cancers and cardiovascular complications and an increase in life expectancy [4,5].

Therefore, the number of surgical procedures in France has increased considerably from 2800 in 1997 to 59,300 in 2016 [6]. Laparoscopic sleeve gastrectomy (LSG) and laparoscopic Roux-en-Y gastric bypass (LRYGB) are the two most performed surgical procedures.

Beyond the surgical act, the most important for the patient is the post-operative follow-up; furthermore, bariatric surgery is performed on young patients with a long-life expectancy underling the need for regular and good quality follow-up to ensure the results in terms of weight loss, the resolution of comorbidities, and good quality of life. According to the HAS guidelines [7], a long-term postoperative multidisciplinary follow-up is recommended because of the risks of weight regain, nutritional deficiencies, psychiatric disorders, and serious medical or surgical complications [8,9,10,11].

Compliance is generally reported to be about 90% in the first year and then decreases drastically to reach 30% in two years and less than 10% in 10 years [12,13,14]. In France, a cohort study showed that the medical follow-up after surgery did not comply with national guidelines [15]. At five years, only 29% of patients had a consultation with a surgeon and 12.4% with an endocrinologist.

This alarming rate of patients with a disrupted follow-up is difficult to determine because of the heterogeneity of the definition. While several authors consider the percentage of follow-up visits, the notion of follow-up disruption is generally defined as a total absence of consultation for a variable period [16,17,18]. In France, a patient is in disruption of follow-up if the patient does not come in a consultation once a year to the referral center, according to the HAS (“Haute Autorité de Santé”) guidelines [7]. Guidelines specified that patients must be followed up four times during the first year, tnotion of follow-up disruption is generally hen twice a year for the second year, and finally yearly during the rest of their life.

Although several factors of disruption follow-up have been described in the literature [16,17,19,20,21,22,23], results about its association with clinical determinants, including weight loss, are heterogeneous [24,25]. Few studies have described the relationship between non-clinical determinants, such as socioeconomic status and geographic accessibility to healthcare [22].

Hence, this study aims to determine the rate and risk factors of disruption follow-up after bariatric surgery in our referral-accredited bariatric center.

## 2. Materials and Methods

### 2.1. Study Design

Data were collected from a prospectively maintained database of patients who underwent laparoscopic bariatric surgery at the University Hospital of Caen, a French specialized and accredited bariatric center, since January 2012 (CNIL 2204611v0). The inclusion criteria involved patients older than 18 years who underwent: (i) primary laparoscopic sleeve gastrectomy (LSG) or laparoscopic Roux-en-Y gastric bypass (LRYGB); (ii) a revisional bariatric surgery only if the initial procedure was completed in our center (i.e., from laparoscopic adjustable gastric band to RYGB and from LSG to RYGB). From January 2012 to December 2019, 1576 consecutive patients were retrospectively analyzed.

### 2.2. Data Collection

#### 2.2.1. Preoperative Data

##### Clinical Data

All relevant data for each patient were collected: age, gender, date of intervention, surgical procedure, preoperative comorbidities related to obesity (diabetes, arterial hypertension, dyslipidemia, and obstructive sleep apnea syndrome), pre-operative body mass index (BMI), preoperative weight loss and American Society of Anaesthesiologists physical status (ASA).

##### Surgery Procedure

Bariatric surgery was proposed to patients in accordance with the French guidelines for bariatric surgery [7]. All indications for primary and revisional bariatric surgeries were approved in an interdisciplinary consensus meeting. All surgical procedures were standardized in our center and were previously described in the literature [26,27].

##### Deprivation Index

Deprivation was assessed using the French version of the European Deprivation Index (EDI, 2011) [28]. The EDI is an aggregated composite index of deprivation in residence, constructed by selecting the fundamental needs associated with both objective and subjective poverty. Patients’ home addresses were geolocated and assigned to an “Ilots Regroupés pour l’Information Statistique” (IRIS) unit, which is the smallest geographical area defined by the Institut National de la Statistique et des Etudes Economiques (INSEE) where census data are available. The French version of the EDI was used to assign a deprivation score to each IRIS. In this study, EDI was used as a continuous variable to assess the impact of social deprivation. Its values, which are centered to zero, vary from negative to positive (−5.332 to 20.522). As the score of EDI is increased (to the most positive values), the social deprivation increases.

##### Geographical Health Accessibility Index

We used the Spatial aCcessibility multiscALar (SCALe) index to estimate accessibility to health care for each patient [29]. It includes data from the Permanent Equipment Base provided by INSEE. Eleven indicators representative of access to primary care (general practitioner, pharmacist, nurses, etc.) were calculated for a residential area and weighted according to the availability of each resource. Combined with the data provided by the health indicators, this index measured health status according to the accessibility of resources. In this study, the SCALe index was used as a continuous variable to assess the impact of geographical health accessibility. Its values, which are centered to zero, vary from negative to positive values (−15.71 to 22.18). As the score of the SCALe index is increased (to the most positive values), the geographic isolation increases.

#### 2.2.2. Postoperative Data

Postoperative weight loss outcomes were assessed using the percent total weight loss (%TWL) as follows [30]:$$\left[\frac{\mathrm{Preoperative}\;\mathrm{weight}\;-\;\mathrm{Weight}\;\mathrm{during}\;\mathrm{follow}\;\mathrm{up}}{\mathrm{Preoperative}\;\mathrm{weight}}\right]\times 100$$

Concerning postoperative data, the exact day of each consultation in our center was collected retrospectively for all patients. The time between each medical visit (including both nutrition and bariatric surgery consultations), the surgery itself, as well as the time between each consultation were calculated. Since some of the surgeries did not correspond exactly to the time recommended, the dates of some consultations were allocated into the following ranges of months: <3, [3–6], [6–12], [12–18], [18–24], [24–36], [36–48], and [48–60]. For each time interval, if the patient had two visits, the average %TWL of those at the boundaries of the interval were assigned, and in all other cases, the minimum %TWL in the interval was selected. Weight loss in patients with revisional surgery was analyzed after the second procedure.

### 2.3. Outcomes

The objective was to assess the influence of clinical determinants on the probability of having a follow-up disruption after bariatric surgery, respecting the definition of the HAS guidelines [7]. However, this can be interpreted in different ways and differs for each patient. Thus, we decided to answer in two ways by modeling first a complete interruption of follow-up and then an irregular follow-up.

### 2.4. Statistical Analysis

For each part of the analysis, describing comparisons were realized using Chi-square’s test for qualitative variables and Wilcoxon’s test for quantitative variables. A *p* < 0.05 was defined as statistically significant.

Our statistical analyses were divided in two parts: on the one hand we analyzed the interruption of follow-up and treated this event with survival analysis. On the other hand, we focused on the irregularity of follow-up. Both of these analyses were complementary because they explored two main parts of the difficulties of post-operative follow-up after BS: The real loss of follow-up was defined as when a patient would never come back to the appointment, while the sequential loss was defined as when a patient would not attend any follow-up appointment over a few month or years but would attend again one day. Due to the repeated data involved in this type of follow-up, we used a multilevel hierarchical analysis to explore these results, as followed down in the manuscript.

In the first part of our results, the interruption of follow-up was modeled as an event that might occur during the postoperative period. Follow-up data were considered and censored with a point date of 31 March 2022. An event of interest was an interruption to the follow-up corresponding a delay between the last visit and the point date superior to 365 days. As the event can occur one time in each patient, the total effective corresponds to the number of patients included with a postoperative consultation at least (*n* = 1576). A Cox model was used to estimate the cumulative rate of follow-up interruption and the hazard ratios with 95% confidence intervals for preoperative variables in univariate analyses. Because postoperative weight loss corresponds to repeated data in the same patient, this was excluded from the analysis. The proportional hazards assumption was verified by the Schoenfeld residual method. In the multivariate model, forced variables such as age, sex, year of intervention, and all variables with p values of less than 0.2 were included.

In the second part, an irregular follow-up was modeled as the time between a consultation with superiors to a point past 365 days. Because of several repeated consultations for each patient, independence between the different measures was not respected, and we considered these data to be longitudinal. The total effective corresponds to the total number of consultations during the first five years for each patient included in the analysis (*n* = 9750). A multilevel logistics model with fixed effects was used to estimate the odds ratio with 95% confidence intervals. All the preoperative variables were analyzed, and the %TWL was also included in univariate analyses. Variables with a *p*-value under 0.2 were included in the multivariate analysis alongside forced variables such as age, sex, and year of intervention. Statistical analyses were performed using SAS 9.3 software (SAS, Cary, NC, USA).

## 3. Results

### 3.1. Results for COX Modeling of Follow Up Interruption

#### 3.1.1. Population Description

From 1 January 2012 to 31 December 2019, bariatric surgery was performed on 1549 patients. The demographic characteristics and preoperative factors of the population were divided into two groups: follow-up interruption (IF+ = 1116 patients) or continued follow-up (IF− = 433 patients) and were compared (Table A1). The proportion of continued follow-up was higher in both women (*p* = 0.0041) and in recent years of intervention (*p* < 0.0001). No significant difference was shown concerning the deprivation index or SCALE score. Patients with lower ASA scores were higher in the group with a follow-up interruption (*p* = 0.0057). Except for OSAS (*p* = 0.0060), no significant difference was described regarding comorbidities or preoperative biometric values.

#### 3.1.2. Follow Up Variables

The mean time of follow-up was 36.94 ± 24.71 months for the population study. With a date point of 31 March 2022, the cumulative rate of follow-up interruption was 72.05%. Table A2 summarizes the cumulative rates until 60 months.

#### 3.1.3. Univariates Analysis

The results are exposed in Table A3. Both male gender (HR = 1.255; CI = 1.092–1.443; *p* = 0.0014) and more recent years of intervention (HR = 1.044; CI = 1.015–1.077; *p* = 0.0035) were associated with a significant increase in the instantaneous risk of follow-up interruption. No significant association was shown concerning socioeconomic deprivation and geographical health accessibility. Revisional surgery (HR = 0.727; CI = 0.600–0.882; *p* = 0.0008) and a higher preoperative BMI (HR = 0.990; CI = 0.981–0.999; *p* = 0.0359) were associated with a low risk of interruption follow-up. No significant association was described for surgical procedures, preoperative weight loss or ASA score, and comorbidities.

#### 3.1.4. Multivariate Analysis

In the final multivariable model (which included age, gender, years of intervention, revisional surgery, preoperative BMI, and ASA score), the independent risk factors for an interruption follow up were male gender (HR = 1.320; CI = 1.147–1.520; *p* = 0.001) and more recent years of intervention (HR = 1.044; CI = 1.013–1.075; *p* = 0.0066). It was found that patients who had revisional surgery had a lower risk of interruption follow-up (HR = 0.751; CI = 0.619–0.912; *p* = 0.0038).

### 3.2. Results for Multilevel Modeling Irregular Follow-Up

#### 3.2.1. Population Description

A total of 9607 consultations were performed on 1549 patients during their first five years of follow-up (Table A4). The consultations were divided into two groups according to periodicity: regular follow-up RF+ (within 365 days) and irregular follow-up RF− (beyond 365 days). No significant difference was shown between the two groups concerning demographic characteristics, socioeconomic deprivation, or geographical health accessibility. Recent interventions (0.0080) and revisional surgery (*p* = 0.0011) were more frequent in the group with regular follow-ups. The means preoperative BMI was lower in the group with an irregular follow-up (*p* = 0.0072). However, no significant difference was found concerning ASA score or comorbidities. The means %TWL was higher in the group of irregular follow-up patients (*p* < 0.001).

#### 3.2.2. Univariates Analysis

The results are summarized in Table A5. Factors including age, gender, surgery, or years of intervention were not associated within the periodicity of follow-up. No significant association was described concerning socioeconomic deprivation and geographical health accessibility. An irregular follow-up had a negative association with revisional surgery (OR = 0.678; CI = 0.537–0.858; *p* = 0.006), as well as a higher pre-operative BMI (OR = 0.980; CI = 0.969–0.991, *p* = 0.006). However, no significant associations were observed regarding ASA score or preoperative comorbidities. Concerning postoperative outcomes, a higher %TWL (OR = 1.038, CI = 1.030–1.045, *p* < 0.0001) was associated with an irregular follow-up.

#### 3.2.3. Multivariate Analysis

In the final model (including age, gender, SCALE score, surgery, years of intervention, revisional surgery, preoperative body mass index, ASA score, and the %TWL), older age (OR 1.013; CI = 1.005–1.021; *p* = 0.0003), male gender (OR = 1.272; CI = 1.047–1.545, *p* = 0.0153) and a higher %TWL (OR = 1.040 CI = 1.032–1.048 *p* < 0.0001) were identified as independent risk factors (Table A5). On the opposite side, a higher preoperative BMI (OR = 0.986; CI = 0.973–0.995; *p* = 0.0340) and revisional surgery (OR = 0.705, CI = 0.543–0.922] were independent protective factors of a discontinued follow-up. An interaction between age and the %TWL was tested, and older patients with lower a %PPT were associated with irregular follow-ups (*p* = 0.042).

## 4. Discussion

In this observational study, the prevalence of follow-up interruptions reached nearly 70% of patients, according to the HAS definitions. This study suggests that the male gender and recent years of intervention are independent risk factors for follow-up interruption. By contrast, patients operated from revisional bariatric surgery had a lower risk of follow-up interruption. Furthermore, older patients of male gender and with a higher weight loss were identified as independent risk factors of irregular follow-ups. On the opposite side, a higher preoperative BMI and revisional surgery were identified as independent protective factors of irregular follow-ups.

The first reports on the metabolic outcomes of bariatric surgery have already described rates below 15% after 10 years [31,32]. A recent review of the literature shows that few studies provide long-term data because of the low rate of follow-up beyond 4 years [33]. Long-term multidisciplinary follow-ups are recommended because poor-quality surveillance is associated with an increased risk of medical, surgical, and/ or psychiatric complications [34]. Our findings are consistent with the literature, and this study reflects the general practice in bariatric centers. The increasing number of surgeries over the years and the cumulative rates of patients with discontinued follow-ups raise concerns among professionals and patients about the feasibility of regular follow-ups [35]. However, publications on this topic from high-volume, specialized university centers probably do not reflect the reality of follow-ups after bariatric surgery and may underestimate follow-ups outside of the center [13]. Moreover, the accurate prevalence is difficult to determine because of the heterogeneity of the definition. While several authors use the percentage of follow-up visits, the notion of “lost-to-follow-up” is usually defined as the interruption of specialized consultations for periods ranging from 6 to 24 months [16,17,18]. In this study, follow-up disruption is described in two different ways: an interrupted follow-up or irregular follow-up, according to the delay outlined by the HAS guidelines [7].

Male gender was identified as the main risk factor for not only follow-up disruption but also irregular follow-ups. Male gender is a risk factor already described in the literature [17,22,23,36].

The instantaneous risk of follow-up interruption was higher during the most recent years of our study. These results could be explained in part by the occurrence of the COVID-19 pandemic, although this hypothesis was not tested in this study. The COVID-19 pandemic has dramatically changed patient follow-up, as recently reported in a French observational study [37]. New strategies such as telemedicine have been developed to improve the follow-up of bariatric surgery patients, but several difficulties have been observed during its implementation [38,39].

In this study, older patients tended to have a more irregular follow-up. This result is inconsistent with the literature, as most studies show that younger age is more often associated with the disruption of postoperative follow-ups [16,17,20,22].

Moreover, an interaction between %PPT and age was significant in our study. In a recent meta-analysis [40], authors found that bariatric surgery achieved weight loss but at a lower rate in younger patients. In addition, postoperative morbidity and mortality rates were higher in older patients after sleeve gastrectomy. Combined with our results, this underlines the importance of follow-up in older patients because of the risk of postoperative complications or insufficient weight loss.

No significant difference was shown concerning non-clinical determinants such as EDI or Scale Score for the interruption or irregularity of follow-up. This result is consistent with the literature. A recent study had found that socioeconomic determinants or geographical health isolation had an impact on the accessibility of bariatric surgery [41]; however, they did not have an impact on the postoperative period, especially for morbidity or mortality. So, it might be possible that the EDI or Scale score only have an impact on preoperative factors in terms of the accessibility of bariatric surgery but not on postoperative periods, such as follow-up.

This study suggests that a higher weight loss determined by the TWL% was an independent risk factor for irregular follow-ups. This finding is currently controversial in the literature. Weight loss is reported to be associated with better adherence, according to a meta-analysis published in 2014 [24]. However, most of the included studies did not use statistical models for longitudinal data, and weight loss was assessed at 12 months. Weight loss after bariatric surgery is nonlinear over time and often reaches a maximum between 12 and 18 months [31]. It is, therefore, possible that the effect of regular follow-up has been overestimated. Lucas et al. [25], using a multilevel linear model, did not find a significant association between the %TWL and attrition. In their study, nearly one-third of patients considered follow-up unnecessary. These results may explain the relationship between weight loss and the regularity of follow-up in our study. Patients who consider their weight loss successful may consider follow-up less necessary and therefore have an irregular follow-up. This study, with an adapted model for longitudinal data and standardized parameters for weight loss, such as the %TWL, is one of the few to find an association between weight loss and post-operative follow-up.

Finally, revisional bariatric surgery was an independent protective factor of follow-up interruptions and irregular follow-ups with a higher preoperative BMI. A higher preoperative BMI, as well as insufficient weight loss, may suggest situations at risk of failure and are, therefore, an indication for revisional bariatric surgery [42,43]. Consequently, patients who are candidates for revisional bariatric surgery enter into a new program with a new multidisciplinary evaluation. The course to revisional surgery is longer and requires a lot of investment on the part of the patient. Thus, revision surgery is associated with regular follow-ups, a higher preoperative BMI, and lower weight loss, which is one of the indicators that support this association.

Interpreting the results of this study, several limitations must be considered. First, this is a retrospective and monocentric study of data collected from a single referral accredited French center. Therefore, a selection bias is possible. The second limitation is related to the %TWL transformation. It involves the absence of weight variation in our time intervals. However, as the consultation time cannot be exactly respected in practice, a real representation of weight loss according to the rhythm of the recommendations is difficult to represent. Finally, missing data about weight in medical records is an integral part of our limitations. However, because of the high number of patients and measurements for each, multiple imputations by a chained equation are discussed [44].

Future studies are needed to accurately assess the causes for the discontinuation of follow-up, which impact the resolution of comorbidities, quality of life, and non-clinical determinants. These are the aims of the upcoming CURATIVE study (CPP N° 2021-A02088-33). In this future study, all patients in our cohort will be re-contacted by a questionnaire to assess several points. The first objective is to determine the presence of follow-up outside our center, to analyze the causes of discontinuation of follow-up, and to assess long-term metabolic outcomes, morbidity, and quality of life.

## 5. Conclusions

Bariatric surgery remains a step in a long course of care in which long-term follow-up is essential. This study suggests that male gender is an independent risk factor for both interruption and irregular follow-up. If recent years are more associated with follow up interruption, a higher weight loss is associated with irregular follow-up. However, it is now a question of determining the reasons for follow-up disruption in these patients in order to facilitate an adherence to postoperative follow-up, and ensure long-term metabolic results, quality of life, and a reduction in complications.

## Data Availability

Not applicable.

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
