# Peer review of "Analysis of the Lack of Follow-Up of Bariatric Surgery Patients: Experience of a Reference Center"

_jcm, 2022, doi:10.3390/jcm11216310_

Round 1

Reviewer 1 Report

This study describes the rate and risk factor of disruption follow-up after bariatric surgery in a bariatric center.

Results suggests that male gender and most recent dates of surgery are two independent risk factors for follow-up interruption. Older age; male gender and higher weight loss were independent risk factors of irregular follow-up and revision bariatric surgery was a protective factor of interruption and irregular follow-up.

 The manuscript is well written. Some issues are the following:

29           It is suggested to describe %TWL abbreviation (Total weight loss)

53            It is suggested to eliminate the full stop and continue with furthermore, for a better understanding of the global idea (54)

57           It is strongly suggested to specify the recommendations for follow-up of the HAS guides and to describe HAS abbreviation

60-64     “Compliance is generally reported to be about 90% in the first year and then decreases drastically to reach 30% at two years and less than 10% at 10 years (12–14). In France, a cohort study had shown that the medical follow-up after surgery did not comply with national guidelines (15). At five years, only 29% of patients had a consultation  with a surgeon and 12.4% with an endocrinologist.”

It is really surprising the low rate of follow up compliance in France. How is it compared with the findings in your study?

 159- 160 The event of interest was an interruption of follow-up corresponding to a delay between the last visit and the point date superior to 365 days.”

It is very difficult to assimilate that any patient undergoing bariatric surgery comply with the event of interest during the first year follow up. Shouldn't the objectives and results of your study be contextualized in that sense?

 382 384, 394, 399 Tables must be properly aligned

 Question. Is there available data on the reason why irregular follow-up subjects withdrew or returned for follow-up visits?

Author Response

Reviewers' comments:

Reviewer #1 (RW1): This study describes the rate and risk factor of disruption follow-up after bariatric surgery in a bariatric center.

Results suggests that male gender and most recent dates of surgery are two independent risk factors for follow-up interruption. Older age; male gender and higher weight loss were independent risk factors of irregular follow-up and revision bariatric surgery was a protective factor of interruption and irregular follow-up.

 The manuscript is well written.

Authors’ response: This positive comments are appreciated, thank you.

Some issues are the following:

29           It is suggested to describe %TWL abbreviation (Total weight loss)

Author’s response : we thank the reviewer for his comment which upgrade our academic work.

Action in the manuscript : we described %TWL in the abstract.

53            It is suggested to eliminate the full stop and continue with furthermore, for a better understanding of the global idea (54)

Author’s response : we thank the reviewer for his comment which upgrade our academic work.

Action in the manuscript : We change the sentence in the manuscript.

57           It is strongly suggested to specify the recommendations for follow-up of the HAS guides and to describe HAS abbreviation

Author’s response : we thank the reviewer for his comment which upgrade our academic work.

Action in the manuscript : HAS recommend to follow patients with one appointment / year post-operatively. We specify it in the manuscript. We change the sentence in the manuscript and added recommendations of HAS for follow-up.

60-64     “Compliance is generally reported to be about 90% in the first year and then decreases drastically to reach 30% at two years and less than 10% at 10 years (12–14). In France, a cohort study had shown that the medical follow-up after surgery did not comply with national guidelines (15). At five years, only 29% of patients had a consultation with a surgeon and 12.4% with an endocrinologist.”

It is really surprising the low rate of follow up compliance in France. How is it compared with the findings in your study?

Author’s response : We thank the reviewer for his comment which upgrade our academic work.

Action in the manuscript : Our results are one of the 37 tertiary referral centers of bariatric surgery in France. They are only a few part of the study of our collegue Jérémie Thereaux who published this national cohort study.

 159- 160 The event of interest was an interruption of follow-up corresponding to a delay between the last visit and the point date superior to 365 days.”

It is very difficult to assimilate that any patient undergoing bariatric surgery comply with the event of interest during the first year follow up. Shouldn't the objectives and results of your study be contextualized in that sense?

Author’s response : we thank the reviewer for his comment which upgrade our academic work.

Action in the manuscript : However, statistical analysis should be performed with an end point date and we have choosen one year according to HAS Guidelines. That is why we contextualized our objectives of our study in that sense.

 382 384, 394, 399 Tables must be properly aligned

 Question. Is there available data on the reason why irregular follow-up subjects withdrew or returned for follow-up visits?

Author’s response : we thank the reviewer for his comment which upgrade our academic work.

Action in the manuscript : We aligned tables. For the question about irregular follow-up subject, this is, without spoilers, the main subject of the next research work about this cohort study.

Reviewer 2 Report

1. The "in text" presented results are fine and give a lot of information
2. Unfortunately there are no graphs added. The tables are hard to read! I think this should be improved.
3. Why only predictive markers are evaluated. It would be interesting to know, if there is a difference in outcome between patient groups.

Author Response

Reviewers' comments:

Reviewer #2 (RW2): 1. The "in text" presented results are fine and give a lot of information

Author’s response This positive comments are appreciated, thank you.

  1. Unfortunately there are no graphs added. The tables are hard to read! I think this should be improved.

Author’s response we thank the reviewer for his comment which upgrade our academic work.

Action in the manuscript : we added a graph in the appendix section and tables were upgraded.

  1. Why only predictive markers are evaluated. It would be interesting to know, if there is a difference in outcome between patient groups.

Author’s response we thank the reviewer for his comment which upgrade our academic work.

Action in the manuscript For the question about outcome between patient groups, this is, without spoilers, the main subject of one of the next researches works about this cohort study.
